# A cross-sectional study examining generalized anxiety disorder among healthcare workers during the COVID-19 pandemic within the Democratic Republic of the Congo

Skylar A. Martin[1], Dalau Mukadi Nkamba[2], Nicole A. Hoff[1], Sydney Merritt[1], Megan Halbrook[1], Sylvia Tangney[1], Nick Ida[1], Gloire Mbaka Onya[3], Armand Mutwadi[2], Kamy Musene[1], Christophe Luhata[4], Didine Kaba[2], Anne W. Rimoin[1]*

1 Department of Epidemiology, Jonathan and Karin Fielding School of Public Health, University of California, Los Angeles, California, United States of America, 2 Kinshasa School of Public Health, University of Kinshasa, Kinshasa, Democratic Republic of the Congo, 3 Malaria Elimination Initiative/ IGHS, Department of Epidemiology and Biostatistics, University of California, San Francisco, California, United States of America, 4 Expanded Program for Immunization, Ministry of Health, Kinshasa, Democratic Republic of the Congo

* arimoin@ucla.edu

## Abstract

Mental health resilience during outbreaks within insecure regions, and the subsequent mental health toll on healthcare workers (HCW), remains a largely unstudied and minimally understood phenomenon. This study examined generalized anxiety disorder (GAD) among healthcare workers (HCWs) and community members in the Democratic Republic of the Congo (DRC) during the COVID-19 pandemic, with a primary focus on the relationship between regional insecurity and GAD. A cross-sectional analysis was conducted using data from 5,622 participants across all 26 DRC provinces collected between February and September 2022. GAD was measured using the GAD-7 screening tool. Multivariate logistic regression was used to assess associations between mild to severe anxiety (GAD score >5) and participant characteristics, including region, housing, and vaccination status. Mild to severe anxiety was identified in 4% of participants. No significant association was found between residence in insecure regions and anxiety. However, being vaccinated against COVID-19 (aOR: 0.68, 95% CI: 0.52–0.89) and having stable housing (aOR: 0.13, 95% CI: 0.05–0.32) were significantly protective. While regional insecurity was not associated with anxiety in this analysis, access to vaccination and housing emerged as key protective factors. These findings highlight urgent directions for intervention and further research on mental health resilience in crisis settings.

**Data availability statement:** The data underlying all findings presented has been uploaded with submission.

**Funding:** Funding for this project is from the U.S. Centers for Disease Control and Prevention, the Task Force for Global Health (TFGH), under grants No. 20214656;2 and 20223122;2 (to AWR). Funders contributed to study design but had no role in data collection, analysis, decision to publish, or preparation of the manuscript. The content of the information does not necessarily reflect the position or the policy of TFGH, and no official endorsement should be inferred.

**Competing interests:** The authors have declared that no competing interests exist.

## Introduction

In the Democratic Republic of the Congo (DRC), insecure regions represent epicenters of persistent conflict and instability. These areas, primarily located in the north-eastern part of the country, endure a complex web of challenges that stem from long standing political and ethnic tensions, further complicated by the presence of numerous armed groups [1]. Among the DRC's twenty-six provinces, five in the east are designated as insecure due to heightened conflict, necessitating increased surveillance by security forces [2,3]. Additionally, the mining of 'conflict minerals' crucial for the global electronics industry is an important economic backdrop for these conflicts [4,5]. Despite the high burden of insecurity, outbreaks, and resource scarcity in the DRC, there is limited research examining how these stressors contribute to generalized anxiety disorder (GAD) among HCWs and community members during the COVID-19 pandemic.

Healthcare workers (HCWs) globally, are disproportionately impacted by stress due to workload, concerns for both patient and personal health, and lack of social support, which is further intensified during outbreaks of infectious disease [6]. Anxiety disorders are the most prevalent class of mental health disorders worldwide, affecting an estimated 301 million people [7]. During public health emergencies, the prevalence of GAD among HCWs has been reported to exceed 30% [8]. During the SARS-CoV-2 (COVID-19) pandemic, HCWs in low-resource settings confronted additional challenges due to significant resource limitations. Occupational risks for HCWs in these settings were increased due to limited personal protective equipment (PPE) resources, lower safety standards of health institutions, and a smaller population of trained HCWs [9,10]. These factors are compounded by the socio-environmental and economic impacts of COVID-19 [11]. Research has shown that relative to their non-healthcare worker counterparts, anxiety and mental health issues have impacted HCWs in other regions globally at a significantly higher magnitude throughout the COVID-19 pandemic [12]. However, in the DRC, HCWs may have been uniquely prepared to confront the COVID-19 pandemic and other emergent diseases after battling multiple Ebola outbreaks, including the second largest recorded Ebola outbreak between 2018 and 2020 [13]. During the Ebola outbreaks, while typically localized, HCWs often received additional training on best practices for infectious disease mitigation. Despite this experience and preparedness, the psychological impact of this novel threat on both HCWs and the general population has not been measured in the DRC, leaving a gap in understanding of the psychological toll of the COVID-19 pandemic in this region. While literature from other conflict-affected countries suggests a relationship between regional insecurity and increased anxiety, few studies have directly examined this intersection within the context of public health crises in sub-Saharan Africa, and none to our knowledge have done so at a national scale in the DRC. As a result, the combined mental health burden of COVID-19, regional conflict, and systemic under-resourcing remains poorly characterized.

Within this complex landscape, HCWs confront a nexus of challenges: managing infectious disease outbreaks, navigating the repercussions of historical and ongoing conflict, and mitigating the environmental and health impacts of mining activities.

These interrelated challenges create a unique backdrop against which the resilience and mental health of HCWs are tested. This analysis explores the interplay of regional insecurity, environmental stressors, and the imminent threat of emerging infectious diseases, such as COVID-19, and their impact on generalized anxiety disorder (GAD) in both health-care workers and community members. We aim to contribute to a nuanced understanding of the mental health challenges faced by the DRC population and offer insights that could inform integrative, evidence-based policy interventions.

## Methods

### Study design and setting

This cross-sectional analysis was nested within a larger longitudinal cohort known as the "COVID-19 Vaccine Rumors Study", conducted across all 26 provinces of the DRC. The primary analysis period for this sub-study was February 28 through September 2, 2022 and focused on generalized anxiety disorder (GAD) as measured by the GAD-7 scale. The parent cohort enrolled participants on a rolling basis from August 19, 2021 through September 1, 2022 and aimed to assess knowledge, attitudes, and rumors regarding COVID-19 and COVID-19 vaccination [14].

### Participants and sampling

A total of 5,622 participants completed the baseline survey and were included in this analysis. Maximum variation sampling was applied by selecting participants across different provinces, occupational categories, and rural/urban settings by recruitment from official lists provided by health and administration authorities, typically at the provincial level. Inclusion criteria included adults (18 years or older) with access to a working phone line and verbal consent to participate. No additional exclusion criteria were applied. Both healthcare workers and community members were included to capture a nationally representative sample and allow for subgroup comparisons. Including both populations was essential to understanding whether occupation modified anxiety risk and to provide adequate statistical power for multivariable modeling.

### Data collection

Trained interviewers completed informed consent procedures where participants were informed of study objectives and procedures prior to enrollment. After the participant provided verbal consent, interviewers documented consent within SurveyCTO in accordance with IRB-approved protocols. Phone interviews were conducted using a structured, tablet-based questionnaire in SurveyCTO and typically took 15–30 minutes to complete. The survey tools were available in French, English, Lingala, Swahili, and Tshiluba, and interviewers would select the most appropriate language based on participant preference. Interviewers were trained through a multi-day protocol that included role-playing, comprehension testing, and guidance on informed consent and culturally sensitive communication and data entry procedures.

### Variables

Primary outcome: The main outcome of interest was anxiety, as measured through the 7-item generalized anxiety disorder scale (GAD-7). GAD-7 is used as a screening tool and severity measure for generalized anxiety disorder, and the score is calculated by assigning scores of 0–3 to responses of 'not at all', 'several days', 'more than half the days', and 'nearly every day', respectively [15]. These individual scores are summed to calculate an overall score with cut-offs of 5 (mild anxiety), 10 (mild anxiety), and 15 (severe anxiety). For this analysis, we dichotomized this score into "mild to severe anxiety" (GAD-7 score >= 5) and "no anxiety" (GAD-7 score < 5) [16].

### Exposures and covariates

Demographic variables included in this analysis included gender, age, highest education level, profession, housing type, urban versus rural residence, comorbidity status and regional insecurity. Five (5) provinces – Bas Uele, Ituri, Tanganyika,

Nord Kivu, and Sud-Kivu – were, based on security classifications, categorized as insecure, and the remaining 21 provinces were classified as secure. Measures related to COVID-19 included vaccination status and Likert scaled-rated questions of participants' perceived risk of infection, the potentiality of the severity of COVID-19 disease, and susceptibility to infection. Other COVID-19 measures included the frequency of conversing with family and friends regarding COVID-19 and whether participants were aware of COVID-19 within their province. Healthcare workers (HCW) were also asked if they believed they had been treated poorly during the COVID-19 pandemic due to their occupation.

## Statistical methods

All statistical analysis was performed using SAS version 9.4 (SAS Institute, Cary, NC). We first performed bivariate logistic regression to examine the association between GAD and each independent variable. Responses with missing data on the primary outcome (GAD-7) were excluded from analysis. Variables with more than 10% missingness were reviewed and sensitivity analyses were conducted to ensure exclusion did not bias final estimates. A multivariate logistic regression model was then applied to assess the joint effects of these variables on GAD. Given the availability of incidence data, odds ratios were used to quantify these relationships. A reduced model was developed using backward variable selection with a significance level of 0.07 to allow for inclusion of potentially meaningful covariates in the context of limited outcome events; the final model included gender, age, housing status, and vaccination status as variables of interest. The significance of the odds ratios was determined using 95% confidence intervals. Adjusted odds ratios (aORs) and 95% confidence intervals were reported.

## Ethical considerations

This assessment was deemed a public health surveillance activity by the Centers for Disease Control (CDC) and the University of California, Los Angeles (UCLA) and was granted a non-research determination; ethical approval was given in-country by the University of Kinshasa, Kinshasa School of Public Health Ethics Committee (IRB: ESP/CE/128/2021). Verbal consent procedures were reviewed and approved by the IRB and were documented by trained staff within the survey software.

## Results

### Descriptive statistics

Among 5,622 participants nationwide, 1,067 (19%) study participants resided in insecure regions and 3,496 (62%) lived in rural regions (Table 1). Male participants made up 75% of the sample and 35% of total participants were between the ages of 35–44. Most participants identified as a HCW (77%). Among the total sample population, 242 (4%) were identified as having mild to severe anxiety via the GAD-7 assessment (>=5 score). In examining GAD scores, public service, essential service, and community leaders represented 15% of participants that were identified as having mild to high anxiety.

Only 4% of HCW participants report being treated poorly during the COVID-19 pandemic due to their HCW status. Beyond profession, factors such as housing (having shelter) and COVID-19 comorbidities were assessed. Just 20 (0.4%) participants were unhoused however a larger proportion of participants with mild to high anxiety (3%) were unhoused, compared to control levels of anxiety. 363 (7%) participants report being immunocompromised or having chronic conditions and 2,999 (53%) report having received at least one dose of COVID-19 vaccination.

### Bivariate model

When examining the association between insecure regions and mild to severe anxiety in a bivariate model, the unadjusted odds ratio (uOR) showed that the odds of mild to severe anxiety are slightly lower but not significant among insecure regions (uOR: 0.97, 95% CI: 0.70, 1.36) as compared to more secure regions (Table 2).

**Table 1.** Study Characteristics Stratified by Exposure (Insecure Region Status) and Outcome (Mild to Severe Anxiety).

| | Mild to Severe Anxiety (n = 242) | | None to Minimal Anxiety (n = 5380) | | Total |
|---|---|---|---|---|---|
| | **Exposed** | **Unexposed** | **Exposed** | **Unexposed** | |
| **Total** | 45 (18.6) | 197 (81.4) | 1022 (19.0) | 4358 (81.0) | 5622 |
| **Sample Characteristic¹** | | | | | |
| **High Security Region** | – | – | – | – | 1067 (18.9) |
| **Gender** | | | | | |
| Male | 33 (73.3) | 139 (70.6) | 765 (74.9) | 3274 (75.1) | 4211 (74.9) |
| Female | 12 (26.7) | 58 (29.4) | 257 (25.2) | 1084 (24.9) | 1412 (25.1) |
| **Age** (categorical) | | | | | |
| 18-24 | 2 (4.4) | 8 (4.1) | 39 (3.8) | 148 (3.4) | 197 (3.5) |
| 25-34 | 8 (17.8) | 31 (15.7) | 282 (27.6) | 841 (19.3) | 1162 (20.7) |
| 35-44 | 17 (37.8) | 69 (35.0) | 374 (36.6) | 1529 (35.1) | 1990 (35.4) |
| 45-54 | 13 (28.9) | 46 (23.4) | 236 (23.1) | 1156 (26.6) | 1451 (25.8) |
| >=55 | 5 (11.1) | 43 (21.8) | 91 (8.9) | 678 (15.6) | 817 (14.6) |
| **Healthcare Worker** | 38 (84.4) | 149 (75.6) | 829 (81.1) | 3320 (76.2) | 4337 (77.1) |
| **Education level** | | | | | |
| No education | 0 (0.0) | 0 (0.0) | 5 (0.5) | 16 (0.4) | 21 (0.4) |
| Less than High School | 7 (15.6) | 12 (6.1) | 149 (14.5) | 352 (8.1) | 520 (9.3) |
| High School Graduate | 17 (37.8) | 55 (27.9) | 345 (33.8) | 1172 (26.9) | 1589 (28.3) |
| Associates Degree | 16 (35.6) | 79 (40.1) | 390 (38.2) | 1968 (45.2) | 2454 (43.6) |
| Bachelor's Degree | 4 (8.9) | 46 (23.4) | 108 (10.6) | 724 (16.6) | 882 (15.7) |
| Advanced Degree | 1 (2.2) | 5 (2.5) | 25 (2.5) | 126 (2.9) | 157 (2.8) |
| **Profession** | | | | | |
| Health Worker | 19 (42.2) | 124 (62.9) | 542 (53.0) | 2612 (59.9) | 3297 (58.6) |
| Community Health Worker | 19 (42.2) | 25 (12.7) | 287 (28.1) | 708 (16.3) | 1040 (18.5) |
| Religious leader | 0 (0.0) | 1 (0.5) | 10 (1.0) | 30 (0.7) | 41 (0.7) |
| Public service/essential service/community leader | 5 (11.1) | 31 (15.7) | 76 (7.4) | 376 (8.6) | 488 (8.7) |
| Other | 2 (4.4) | 13 (6.6) | 80 (7.8) | 486 (11.2) | 581 (10.3) |
| Not currently in paid work | 0 (0.0) | 3 (1.5) | 27 (2.6) | 146 (3.4) | 176 (3.1) |
| **Residence type** | | | | | |
| House/Apartment | 43 (95.6) | 193 (98.0) | 1018 (99.6) | 4344 (99.7) | 5599 (99.6) |
| Unhoused/Unstable housing | 2 (4.4) | 4 (2.0) | 2 (0.2) | 12 (0.3) | 20 (0.4) |
| Other | 0 (0.0) | 0 (0) | 2 (0.2) | 2 (0.05) | 4 (0.1) |
| **Rural** | 39 (86.7) | 119 (60.4) | 721 (70.6) | 2617 (60.1) | 3496 (62.2) |
| **Comorbidity status** | 1 (2.2) | 19 (9.6) | 44 (4.3) | 299 (6.9) | 363 (6.5) |
| **Frequent Conversations with Family and Friends** | 30 (66.7) | 118 (59.9) | 556 (54.4) | 2627 (60.3) | 3331 (59.2) |
| **Treated Poorly During COVID-19 Pandemic due to HCW Status** | 0 (0.0) | 7 (4.7) | 19 (2.3) | 145 (4.4) | 171 (3.9) |
| **Aware of COVID-19 Within Province** | 43 (100.0) | 172 (92.5) | 888 (95.3) | 3770 (93.7) | 4874 (94.0) |
| **Vaccinated Against COVID-19** | 22 (48.9) | 82 (42.5) | 545 (53.3) | 2349 (53.9) | 2999 (53.4) |
| **High Risk of COVID-19 Infection** | 16 (35.6) | 79 (40.1) | 399 (39.0) | 1774 (40.7) | 2268 (40.3) |
| **High Severity of COVID-19 Disease** | 13 (28.9) | 65 (33.0) | 272 (26.6) | 1533 (35.2) | 1883 (33.5) |
| **High Susceptibility to COVID-19 Infection** | 24 (53.3) | 68 (34.5) | 557 (54.5) | 1731 (39.7) | 2381 (42.3) |

¹Reported as n (%) unless otherwise specified.

**Table 2. Odds Ratios for Mild to Severe Anxiety and Study Characteristics.**

| | Bi-variate | Multivariate regression - Full Model |
|---|---|---|
| | UOR (95% CI) | AOR (95% CI) |
| **Sample characteristic¹** | | |
| **Regional Security** | | |
| Insecure Region | 0.97 (0.70, 1.36) | 1.00 (0.70, 1.43) |
| Low-Security Region | 1.00 | 1.00 |
| **Profession** | | |
| Healthcare worker | 1.01 (0.74, 1.37) | 1.11 (0.79, 1.56) |
| Non-Healthcare worker | 1.00 | 1.00 |
| **Gender** | | |
| Male | 0.82 (0.61, 1.09) | 1.35 (1.00, 1.82) |
| Female | 1.00 | 1.00 |
| **Age** | | |
| Under 55 Years of Age | **0.67 (0.49, 0.93)** | **1.53 (1.07, 2.19)** |
| Over 55 Years of Age | 1.00 | 1.00 |
| **Region** | | |
| Rural | 1.15 (0.88, 1.51) | 1.16 (0.87, 1.54) |
| Urban | 1.00 | 1.00 |
| **Education** | | |
| Any College | 1.01 (0.78, 1.32) | 1.08 (0.80, 1.45) |
| High School and Less | 1.00 | 1.00 |
| **Housing Status** | | |
| Resides in house or apartment | **0.11 (0.07. 0.25)** | **0.12 (0.05, 0.31)** |
| Homeless or Other | 1.00 | 1.00 |
| **Comorbidity Status** | | |
| Comorbidity | 1.32 (0.83, 2.12) | 0.90 (0.51, 1.59) |
| No Comorbidity | 1.00 | 1.00 |
| **Frequent Conversations with Family and Friends** | | |
| Yes | 1.09 (0.83, 1.42) | 1.17 (0.88, 1.56) |
| No | 1.00 | 1.00 |
| **Aware of COVID-19 Within Province** | | |
| Yes | 0.98 (0.56, 1.70) | 1.00 (0.57, 1.77) |
| No | 1.00 | 1.00 |
| **Vaccinated Against COVID-19** | | |
| Yes | **0.67 (0.51, 0.87)** | **0.70 (0.53, 0.93)** |
| No | 1.00 | 1.00 |
| **High Risk of COVID-19 Infection** | | |
| Yes | 0.95 (0.73, 1.24) | 0.97 (0.85, 1.11) |
| No | 1.00 | 1.00 |
| **Perceived High Severity of COVID-19 Disease** | | |
| Yes | 0.94 (0.72, 1.24) | 1.07 (0.94, 1.20) |
| No | 1.00 | 1.00 |
| **High Susceptibility to COVID-19 Infection** | | |
| Yes | 0.83 (0.64, 1.08) | 1.02 (0.90, 1.16) |
| No | 1.00 | 1.00 |

¹Reported as OR (95% CI) unless otherwise stated.

²Bolded values indicate significance.

Vaccination significantly decreased the odds of mild to severe anxiety (uOR: 0.67, 95% CI: 0.51, 0.87). Younger participants (under the age of 55) had lower odds of anxiety as compared to older participants (uOR: 0.67, 95% CI: 0.49, 0.93).

**Full multivariate model**

The full multivariate regression model indicated no significant association between insecure regions and mild to severe anxiety within this sample (aOR: 1.00, 95% CI: 0.70, 1.43) (Table 2). HCWs had a non-significant, 11% increase in odds of mild to severe anxiety as compared to non-healthcare workers (aOR: 1.11, 95% CI: 0.79, 1.56). This model indicated a significant association between housing status and anxiety. While unhoused participants made up a very small portion of the total population (0.4%), being housed was highly protective for mild to severe anxiety (aOR: 0.12, 95% CI: 0.05, 0.31). In this model, younger participants had increased odds of mild to high anxiety (aOR: 1.53, 95% CI: 1.07, 2.19). Male participants had a 32% increase in the odds of being identified as having mild to severe anxiety as compared to their female participants (aOR: 1.32, 95% CI: 1.00, 1.82). Vaccination significantly decreased the odds of mild to severe anxiety; vaccinated participants were 30% less likely to be identified as having mild to severe anxiety as compared to unvaccinated participants (aOR: 0.70, 95% CI: 0.53, 0.93). This is consistent with the crude odds ratio.

**Reduced model**

Following the reduction of the full multivariate model (Fig 1), the odds of mild to severe anxiety were 1.51 times higher among participants under 55 years of age (aOR: 1.51, 95% CI: 1.07, 2.13). Men were more likely to experience mild to severe anxiety, though this association was borderline significant (aOR: 1.33, 95% CI: 0.99, 1.80). Being vaccinated against COVID-19 was significantly associated with reduced odds of mild to severe anxiety (aOR: 0.68, 95% CI: 0.52,

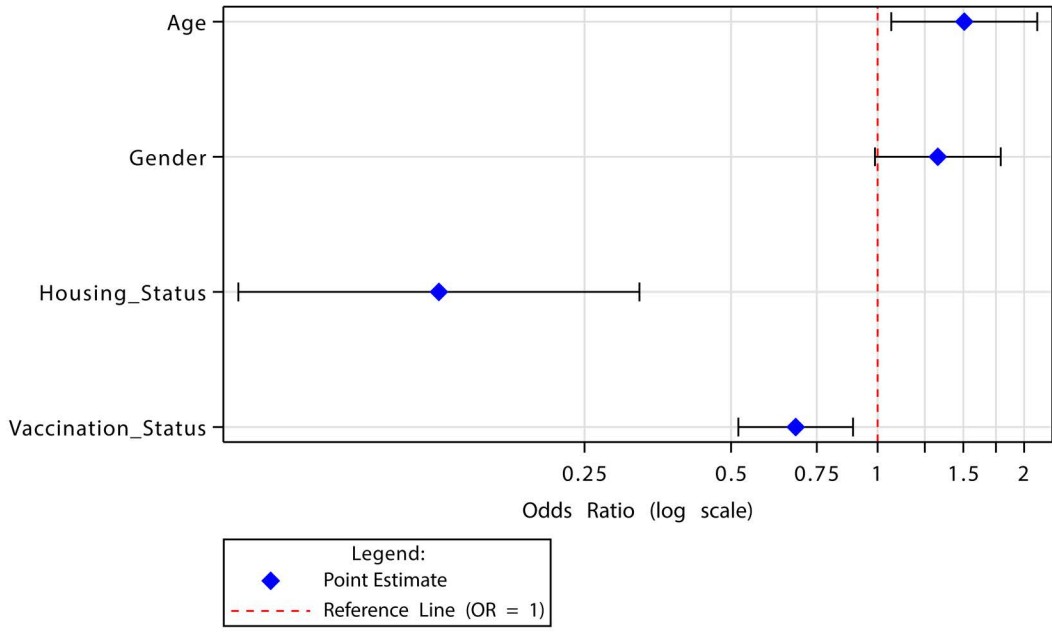

**Fig 1. Forest Plot of Reduced Logistic Regression Model.**

0.89). A strong negative association was established between residing in a house or apartment and mild to severe anxiety (aOR: 0.13, 95% CI: (0.05, 0.32) Table 3.

## Temporal and geographic patterns

We observed spikes in anxiety prevalence during the months of February and July (Fig 2). While the data used for this analysis is cross-sectional and does not include follow-up data, these spikes in anxiety frequency may be due to increases in unrest during these periods of time (2). In July, both secure and insecure provinces had an uptick in the proportion of anxiety

**Table 3. Multivariate Regression Odds Ratios for Mild to Severe Anxiety - Reduced Model.**

| Sample characteristic[1] | AOR (95% CI) |
|---|---|
| **Gender** | |
| Male | 1.33 (0.99, 1.80) |
| Female | 1.00 |
| **Age** | |
| Under 55 Years of Age | **1.51 (1.07, 2.13)** |
| Over 55 Years of Age | 1.00 |
| **Housing Status** | |
| Resides in house or apartment | **0.13 (0.05, 0.32)** |
| Homeless or Other | 1.00 |
| **Vaccinated Against COVID-19** | |
| Yes | **0.68 (0.52, 0.89)** |
| No | 1.00 |

[1]Reported as OR (95% CI) unless otherwise stated.

[2]Bolded values indicate significance.

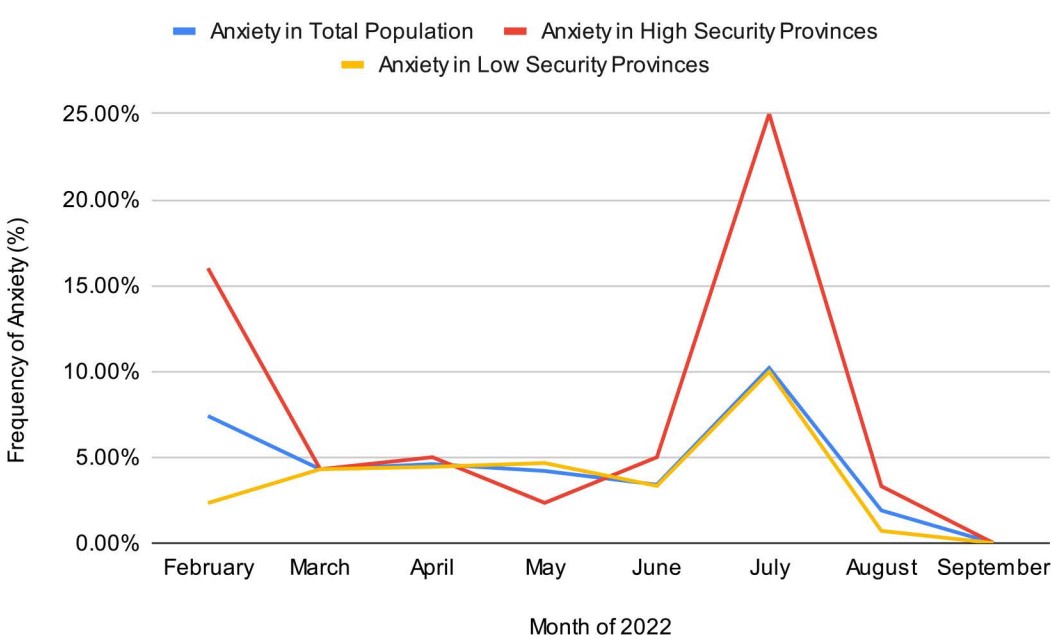

**Fig 2. Prevalence of Moderate to Severe Anxiety by Month of Study.**

among participants, whereas in February, the spike was only within insecure provinces. These values are not stratified by location, which may also play a role in the differences in the number of participants with anxiety at different time points, as different provinces were surveyed. Of note, the four provinces with highest GAD are Sankuru, Tanganyika, Maniema, and Kasai, respectively (Fig 3). Only Tanganyika is classified as insecure, while the other three are considered secure regions. However, these provinces may have areas which experience localized insecurity compared to province-wide insecurity.

Maps were created using ArcGIS software by Esri. ArcGIS and ArcMap are the intellectual property of Esri and are used herein under license.

## Discussion

In this analysis, we explored levels of generalized anxiety disorder (GAD) within the DRC among HCWs and community members, amidst the challenges of the COVID-19 pandemic. In particular, we explored GAD in insecure regions, namely the north-eastern provinces of DRC. Our results revealed a complex interplay of factors influencing anxiety, with notable

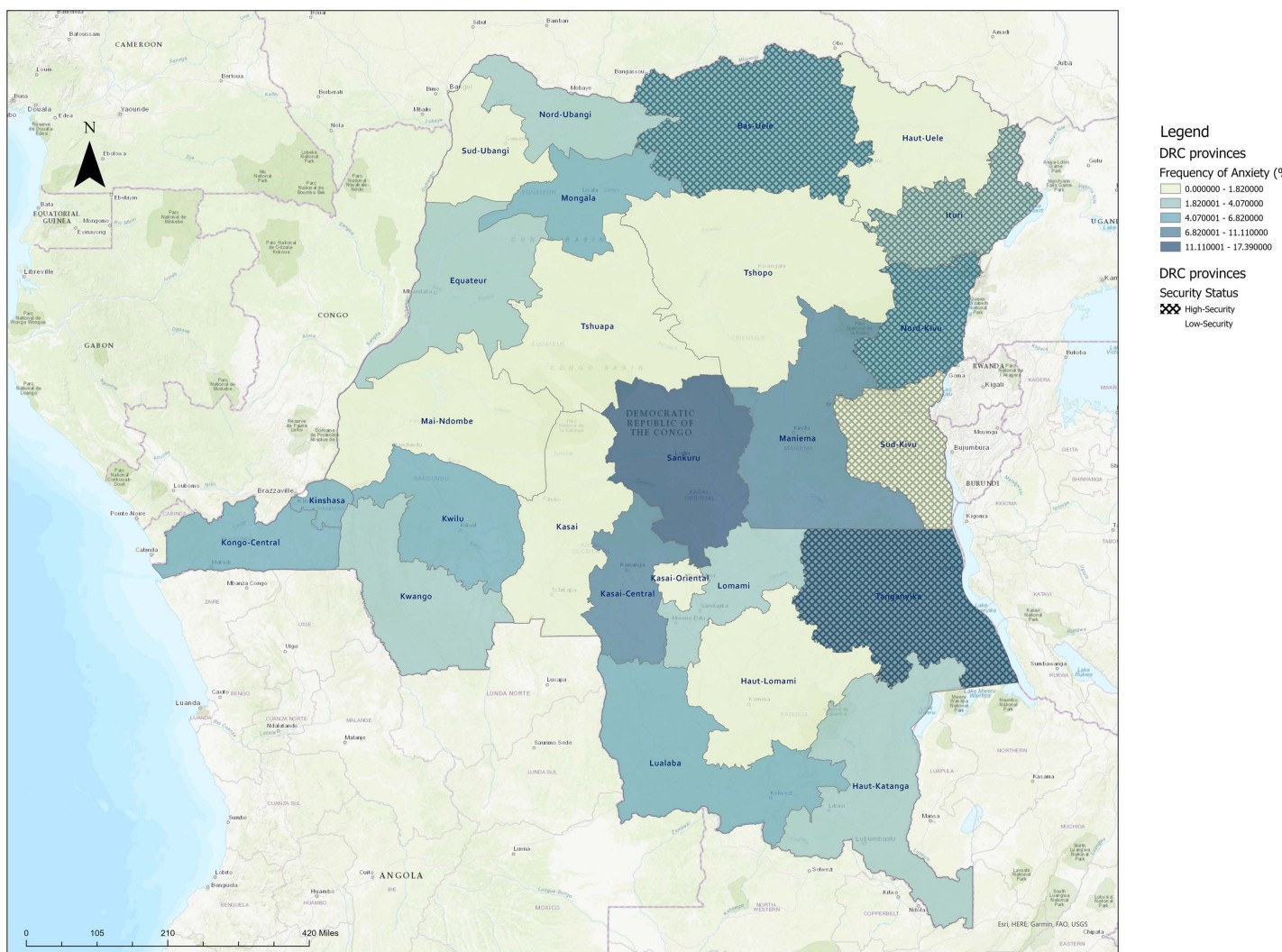

**Fig 3. Prevalence of Anxiety across the Democratic Republic of the Congo by Province and Security Status.**

deviations from existing literature. This finding may reflect a gap in available research, highlighting the need for future work to better capture the mental health landscape in this region which could have implications for policy and practice, and open new directions for future research. While this analysis found no association between insecure regions and mild to severe anxiety, it raises questions about whether current tools or frameworks, including the GAD-7, are adequate to capture mental health outcomes in conflict zones. Previous literature established a link between generalized anxiety and conflict zones [17], but our results suggest this relationship may not be straightforward. This discrepancy could be driven by biases related to the use of phone interviews or the systematic error arising from use of the GAD-7, a measure validated elsewhere but not specifically for the DRC, as well as very small sample sizes in some strata. Further investigation into the resilience mechanisms at play within these communities is necessary to better understand this association and the mental health status of populations in insecure regions.

Participants included in this analysis appear to be particularly resilient based on response classifications, especially when considering the majority of participants were HCW. Global anxiety prevalence as of 2020 is estimated to be 41%, much higher than the measured prevalence of anxiety (4%) in this study [18]. HCWs in our population also did not report higher levels of anxiety compared with their non-HCW counterparts. This finding, while inconsistent with global data, may reflect unique resilience of HCWs in the DRC, particularly given their extensive experience with previous infectious disease outbreaks such as Ebolavirus disease. It is possible that factors including prior experience and extensive training play a role in how HCW perceive and cope with stress during new outbreaks, but further research is needed to explore this in greater depth.

Our findings underscore the vital roles of vaccination and housing in mitigating anxiety, within the context of a novel infectious disease outbreak. We found that participants who received COVID-19 vaccinations or lived in a house or apartment, were markedly less likely to experience anxiety, reinforcing the recognized significance of these elements in fostering mental health resilience [19]. Of note, the analysis presented a striking 88% reduction in anxiety for individuals with housing, compared to those lacking housing, underscoring the fundamental need for safe housing in supporting mental health and resilience. Furthermore, the protective effect of vaccination against anxiety was evident, with vaccinated individuals showing a 30% reduction in odds of anxiety compared to unvaccinated participants. This finding is especially relevant in the DRC, where vaccine hesitancy for new vaccines can pose a major challenge, emphasizing the imperative for ongoing efforts in vaccination education and distribution to enhance both physical and mental health outcomes [20].

Male and younger participants had higher odds of mild to severe anxiety; however, these associations may reflect confounding or effect modification, as we observed a reversal in the direction of association after adjustment for age and gender. Further analysis of these variables is needed to more deeply understand how gender and age impact the incidence of generalized anxiety. A weak positive association was observed between higher education and mild to severe anxiety, indicating there may exist a dose-response relationship, where generalized anxiety may increase as education attainment increases. Existing literature does not support this relationship, and overall, the relationship between education and anxiety remains unclear [21]. In this population, it may be that lower educational attainment may result in individuals being unaware of the risks of COVID-19 and thus less likely to feel stress or anxiety related to the pandemic. However, more comprehensive research is needed to explore this potential relationship, especially considering the complex dynamics of education, health literacy, and anxiety in public health crises.

Notably, the study's proxy variables for emotional stress related to COVID-19 showed limited or no association with mild to severe anxiety. Self-belief in high risk of infection, expectations of severe disease upon infection, and high susceptibility to COVID-19 infection had weak or null associations with mild to severe anxiety within this population. This may reflect the time frame of this analysis, where the perceived threat of COVID-19 infection had already diminished in the DRC, thus weakening its emotional impact. Perhaps more surprisingly, the proxy variable for social support, speaking with family and friends about COVID-19 frequently, showed a non-significant increase in the odds of anxiety. These findings suggest that additional research is required to refine mental health measures in regions facing ongoing public health crises and conflict, including those now affected by the Mpox epidemic [22].

While this population provides a national sample, there are a number of limitations, which could have an impact on generalizability of these findings. The first is the selected time frame. While it is valuable to understand the anxiety levels of this population at any time frame, evaluating the COVID-19 pandemic's impact on GAD would be more useful if this study was conducted with data points from the onset of the outbreak. Additionally, while we observed isolated increases in anxiety during the months of February and July, our study design does not allow us to definitively determine whether these reflect COVID-19 waves, local unrest, or other temporal factors. Future longitudinal designs could help clarify whether such spikes represent transient stress responses or more systematic patterns. This study began collecting data in August 2021 but later added questions related to GAD in February 2022, giving limited insight into the outbreaks' initial impacts on healthcare worker mental health and resiliency. Additionally, this survey was limited to participants with a working phone, which likely represents a more educated and higher socioeconomic status population compared to the general population, who may have less anxiety compared to those without phones or mobile coverage. Another potential bias of this work is the use of the GAD-7. While the GAD-7 has been broadly used and is validated within high-income regions, it has not been validated in the DRC. Further research is needed in validating culturally appropriate measures of mental health in low- and middle-income countries and the DRC, specifically [23]. This study only assessed generalized anxiety disorder (GAD) using the GAD-7 screening tool and did not include other types of anxiety disorders such as panic disorder, social anxiety disorder, or specific phobias. This limitation was due to the structure of the parent study, which did not focus on mental health as a primary aim. The inclusion of GAD-7 was a supplementary effort to explore mental health trends. As a result, our findings reflect only one dimension of anxiety and may underestimate the broader burden of anxiety disorders in this population. Finally, while interviewers were trained, all responses were self-reported through the phone - thus interviewer bias and unclear communication lines during the survey may have indicated a population that had less anxiety than the general population.

Only 19% of the total sample is made up of participants residing in high-security regions, the exposure of interest. Assessing the impact of this exposure is logistically challenging and has limited precision due to the relatively small sample size. Further, we used proxy variables to assess participant resiliency, rather than previously accepted resiliency scores. Other studies that examine resiliency typically use a combination of anxiety, depression, and overall mental well-being as a summed score to evaluate the population's resiliency [24]. This study could only examine anxiety, rather than a summed resiliency score, which gives limited insight into participants' overall mental health. However, inclusion of the full resiliency questionnaire may be challenging to administer within the DRC as clinical mental health is not yet widely available or accepted [25]. By expanding research on this topic, best practices for mental health education and interventions can be developed, relative to the region and population [26].

It is well established that HCWs are at greater risk of experiencing mental health challenges, particularly when faced with an infectious disease outbreak. As HCWs are an important resource, an enhanced focus on their long-term mental health is needed to create strong healthcare systems to endure emerging public health risks. With the increasing frequency of epidemics in the DRC, such as the ongoing Mpox outbreak, this focus on HCW mental health is even more critical. Outbreaks tend to increase workload and overall stress, especially when dealing with a complex virus like COVID-19. Creating a resilient community, both within and outside the healthcare industry, is important for long-term health and success in the region. Access to housing serves as a protective factor against anxiety, promotes long-term improvements in mental health, and plays a crucial role in maintaining physical well-being during infectious disease outbreaks. Investment in housing resources for low-income citizens, as well as financial support for the development of low-cost housing, may foster community-wide resilience.

While this study did not reveal clear associations between anxiety and conflict zones, it underscores the complex and layered nature of psychological well-being in areas facing both conflict and public health crises. As one of the first national-level analyses of GAD in the DRC during the COVID-19 pandemic, this study highlights the protective roles of housing and vaccination, and the urgent need for context-specific mental health assessments. Our findings point to critical directions for future research, particularly in refining culturally sensitive, validated tools that can more accurately capture

mental health outcomes specific to low-resource and conflict-affected settings. Longitudinal studies are also needed to follow mental health trajectories over time, offering deeper insights into resilience and vulnerability as persistent threats – such as displacement, disease outbreaks, and environmental disruption – continue to shape the region. By addressing these aspects, we can better support healthcare workers and affected communities through more effective public health responses, ultimately informing policies and interventions that foster robust mental health resilience in insecure regions.

## Supporting information

**S1 Data. De-identified dataset used for the analysis, including participant demographics, anxiety scores, and regional classifications.**
(XLSX)

**S1 Checklist. Inclusivity in global research questionnaire.**
(DOCX)

## Acknowledgments

We thank Brooke Aksnes, Emma Wray Aberle-Grasse and Reena Doshi for their continued support of this ongoing research endeavor.

## Author contributions

**Conceptualization:** Dalau Mukadi Nkamba, Nicole A. Hoff, Sydney Merritt, Megan Halbrook, Sylvia Tangney, Kamy Musene, Christophe Luhata, Anne W. Rimoin.

**Data curation:** Skylar A Martin, Nicole A. Hoff, Sydney Merritt, Megan Halbrook, Sylvia Tangney, Nick Ida, Gloire Mbaka Onya, Armand Mutwadi, Kamy Musene, Christophe Luhata.

**Formal analysis:** Skylar A Martin, Dalau Mukadi Nkamba, Nicole A. Hoff, Sylvia Tangney, Nick Ida.

**Funding acquisition:** Didine Kaba, Anne W. Rimoin.

**Investigation:** Dalau Mukadi Nkamba, Nicole A. Hoff, Sydney Merritt, Sylvia Tangney, Gloire Mbaka Onya, Armand Mutwadi, Kamy Musene, Christophe Luhata, Didine Kaba, Anne W. Rimoin.

**Methodology:** Skylar A Martin, Nicole A. Hoff, Megan Halbrook, Sylvia Tangney, Nick Ida, Gloire Mbaka Onya, Armand Mutwadi, Kamy Musene, Christophe Luhata, Didine Kaba.

**Project administration:** Armand Mutwadi, Didine Kaba, Anne W. Rimoin.

**Supervision:** Sylvia Tangney, Gloire Mbaka Onya, Didine Kaba, Anne W. Rimoin.

**Validation:** Skylar A Martin, Dalau Mukadi Nkamba.

**Visualization:** Skylar A Martin.

**Writing – original draft:** Skylar A Martin, Nicole A. Hoff.

**Writing – review & editing:** Skylar A Martin, Dalau Mukadi Nkamba, Nicole A. Hoff, Sydney Merritt, Megan Halbrook, Sylvia Tangney, Nick Ida, Gloire Mbaka Onya, Armand Mutwadi, Kamy Musene, Christophe Luhata, Didine Kaba, Anne W. Rimoin.

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
