## [Decision Letter · Decision Letter 0]

1 Apr 2025

PMEN-D-24-00557

A Cross-Sectional Study Examining Anxiety Levels of Healthcare Workers and Community Members Within the Democratic Republic of the Congo

PLOS Mental Health

Dear Dr. Rimoin,

Thank you for submitting your manuscript to PLOS Mental Health. I am very sorry for the delay in reaching a decision - this has been due to difficulties securing reviewers. After careful consideration of the reviewer reports, which we have now obtained, we feel that it has merit but does not yet fully meet PLOS Mental Health’s publication criteria as it currently stands. Therefore, we invite you to submit a revised version of the manuscript that addresses the points raised during the review process.

Please ensure that you address all of the comments raised by the reviewers and please feel free to reach out if you have any questions.

We look forward to receiving your revised manuscript.

Kind regards,

Karli Montague-Cardoso

Executive Editor

PLOS Mental Health

Journal Requirements:

1. Please include a complete copy of PLOS’ questionnaire on inclusivity in global research in your revised manuscript. Our policy for research in this area aims to improve transparency in the reporting of research performed outside of researchers’ own country or community. The policy applies to researchers who have travelled to a different country to conduct research, research with Indigenous populations or their lands, and research on cultural artefacts. The questionnaire can also be requested at the journal’s discretion for any other submissions, even if these conditions are not met.  Please find more information on the policy and a link to download a blank copy of the questionnaire here: https://journals.plos.org/plosmentalhealth/s/best-practices-in-research-reporting. Please upload a completed version of your questionnaire as Supporting Information when you resubmit your manuscript.

2. In the ethics statement in the Methods, you have specified that verbal consent was obtained. Please provide additional details regarding how this consent was documented and witnessed, and state whether this was approved by the IRB.

3. We do not publish any copyright or trademark symbols that usually accompany proprietary names, eg (R), (C), or TM (e.g. next to drug or reagent names). Please remove all instances of trademark/copyright symbols throughout the text, including ® and ™ on page 17.

Additional Editor Comments (if provided):

Reviewers' comments:

Reviewer's Responses to Questions

**Comments to the Author**

1. Does this manuscript meet PLOS Mental Health’s publication criteria ? Is the manuscript technically sound, and do the data support the conclusions? The manuscript must describe methodologically and ethically rigorous research with conclusions that are appropriately drawn based on the data presented.

Reviewer #1: Partly

Reviewer #2: Yes

2. Has the statistical analysis been performed appropriately and rigorously?

Reviewer #1: Yes

Reviewer #2: Yes

3. Have the authors made all data underlying the findings in their manuscript fully available (please refer to the Data Availability Statement at the start of the manuscript PDF file)?

Reviewer #1: Yes

Reviewer #2: No

4. Is the manuscript presented in an intelligible fashion and written in standard English?

Reviewer #1: Yes

Reviewer #2: Yes

5. Review Comments to the Author

Reviewer #1: Manuscript ID: PMEN-D-24-00557–titled: “A Cross-Sectional Study Examining Anxiety Levels of Healthcare Workers and Community Members Within the Democratic Republic of the Congo”

I read with interest this manuscript, which is very interesting and presents original findings, to the best of my knowledge. First, I want to congratulate the authors on the quality of their work. This manuscript has the merit of being published soon if the authors consider certain aspects to improve the quality of its content.

General remarks and advice

After careful reading of this manuscript, I understand that this study sought to estimate the prevalence of generalized anxiety disorders among healthcare workers (accounted for 77% of the sample) who were confronted with the Covid-19 pandemic, then to determine the anxiety levels in those who suffered from them and to identify the associated factors with the risk of developing these anxiety disorders. I understand that the general population was also included in the broad Covid vaccination study. However, including the general population (community members) in this highly relevant study seems to dilute its focus.

If the authors used the “7-item generalized anxiety disorder (GAD-7) scale” as a data collection tool, I suggest that, from the title right through to the body of the manuscript, they clearly refer to “generalized anxiety disorder” and not primarily to “anxiety level”.

It is also important to emphasize the key concept “COVID-19 pandemic” in the title. Unless the authors find it inappropriate, wording such as “A Cross-Sectional Study Examining Generalized Anxiety Disorders Among Healthcare Workers During The COVID-19 Pandemic Within the Democratic Republic of the Congo” may be clear to potential readers.

I would suggest that the authors do not place too much emphasis in this analysis on the association between “politically insecure regions and anxiety levels”. This aspect is more complex than it is easy to address in a cross-sectional study such as this.

Some sections of the manuscript appear incomplete. I advise the authors to use the STROBE guidelines to complete all missing aspects, particularly in the Methods and Discussion sections.

Section-by-Section Specific Comments

Introduction

The research problem is clearly stated in paragraphs 2 and 3. However, what is missing is a description of the burden (epidemiological burden, economic burden, social burden, etc.) of anxiety disorders, supported by the evidence, to back up this stated problem. It is important to expand on the state of the art with statistical data.

The purpose could be a little clearer. Sometimes the authors state as follows: “This analysis explores the interplay of regional insecurity, environmental degradation, and the imminent threat of emerging infectious diseases, such as COVID-19 impacts anxiety levels” (Lines 90-92), sometimes as follows: “We aim to contribute to a nuanced understanding of the mental health challenges faced by the DRC population and offer insights, which could inform integrative policy interventions.” (Lines 92-94).

Methods

The Methods section contains very relevant information describing the research approach adopted. However, this information seems to be jumbled together, creating some confusion for the reader. To make this section much clearer for potential readers, I suggest subdividing it into the following sub-sections: study design; study setting; population and sampling; data collection methods and tools; data collection procedure; data analysis; data management and ethics, etc. Please use the STROBE guidelines to further clarify this section.

The authors’ state: “phone interviews using a structured tablet-based questionnaire in SurveyCTO were conducted and typically took 15 to 30 minutes to complete. (Lines 106-107). Does this suggest that every participant in the study had a telephone? What then were all the inclusion and exclusion criteria for the study?

Apart from generalized anxiety disorders, which were measured using the 7-item generalized anxiety disorder scale (GAD-7), what other tools did the authors use to collect other data such as those mentioned on lines 110-111? If possible, it will be important to describe them very briefly as they have not been added as supplementary files.

Results

In the following finding: “The full multivariate regression model indicated no significant association between insecure regions and mild to severe anxiety within this sample (aOR: 1.00, 95% CI: 0.70, 1.43)” (Lines 169-170), the authors note that there is no association between “insecure regions” and “anxiety disorder”, whereas this association is found elsewhere. Is this a problem of study design, information bias, measurement bias, etc.? It is important to discuss this result in depth and to call for further in-depth studies in this context.

Normally, this result (aOR: 1.33, 95% CI: 0.99, 1.80) suggests an absence of association between gender and anxiety risk. Figure 3 shows this clearly. If you believe that there is indeed an association, it is necessary to round the lower limit of the confidence interval from 0.99 to 1.00. This would avoid passing through 1 in the confidence interval between the lower and upper limits. If you agree with me, please also amend the abstract.

The authors observed an increase in the prevalence of anxiety disorders at a given time: “We observed spikes in anxiety prevalence during the months of February and July” (Line 196). Did this correspond to a wave (outbreak) of COVID-19 cases, or is there another explanation? If possible, discuss this increase in the prevalence.

Discussion

An anxiety disorder prevalence 10 times lower than the international literature, yet in the context of emerging COVID-19 and insecurity? This should be widely discussed.

There are at least 6 types of anxiety disorder: panic disorder, agoraphobia, generalized anxiety disorder (GAD), social anxiety disorder, specific phobia, and separation anxiety disorder. In this study, the authors dealt with only one type: GAD. The Limitations of the study sub-section would mention the fact that the study was limited to this type of disorder and would justify why other types were not considered.

Reviewer #2: Thank you for giving me the opportunity to review this work. Please find a summary of my review

Section-wise Review & Suggested Edits

1. Title & Abstract

• The title should be concise and accurately reflect the core argument or findings of the paper. Consider making it more engaging.

• The abstract should provide a clearer summary of the research question, methodology key findings, and significance. If possible, make it structured.

2. Introduction

Strengths:

• The introduction provides background on the topic.

Areas for Improvement:

• The problem statement should be clearer. Why is this topic important?

• The research gap should be explicitly stated—how does this article contribute to the existing body of knowledge?

Suggested Edits:

• Strengthen the opening by clearly stating the research question or main argument.

3. Literature Review / Background

Strengths:

• Covers relevant sources related to the topic.

Areas for Improvement:

• Some sections lack sufficient citations. Claims should be backed by evidence from existing studies.

• The connection between previous research and the current study should be clearer.

Suggested Edits:

• Add more references where needed.

• Clearly state how the reviewed literature informs the research.

4. Methodology (If Applicable)

Strengths:

• Describes the approach used for data collection or analysis.

Areas for Improvement:

• Lacks detail on why a particular methodology was chosen.

Suggested Edits:

• Provide a rationale for the chosen research method.

• Ensure reproducibility by adding sufficient methodological details.

5. Results & Discussion

Strengths:

• The discussion touches on key findings.

Areas for Improvement:

• The results should be explicitly stated before discussion.

• The discussion should connect findings to existing literature—do they support or contradict previous studies?

• Consider structuring this section more clearly.

Suggested Edits:

• Clearly separate results from discussion for clarity.

• Use subheadings to organize findings logically.

6. Conclusion & Recommendations

Strengths:

• Summarizes key points.

Areas for Improvement:

• The conclusion should more explicitly state the study's contribution.

• Practical implications or future research directions could be discussed.

Suggested Edits:

• Reinforce the significance of the findings.

• Suggest future research directions to strengthen impact.

7. Language & Formatting

Areas for Improvement:

• Some grammatical errors and obstinate phrasing should be revised.

• Ensure consistency in referencing style (APA, MLA, etc.).

Suggested Edits:

• Proofread for grammatical and syntactical errors.

• Use clear and precise academic language.

6. PLOS authors have the option to publish the peer review history of their article (what does this mean? ). If published, this will include your full peer review and any attached files.

**Do you want your identity to be public for this peer review?** For information about this choice, including consent withdrawal, please see our Privacy Policy .

Reviewer #1: **Yes: ** Erick Mukala Mayoyo

Reviewer #2: **Yes: ** Dr David Onchonga

---

## [Decision Letter · Decision Letter 1]

1 Jul 2025

A Cross-Sectional Study Examining Generalized Anxiety Disorder Among Healthcare Workers During the COVID-19 Pandemic Within the Democratic Republic of the Congo

PMEN-D-24-00557R1

Dear Dr. Rimoin,

We are pleased to inform you that your manuscript 'A Cross-Sectional Study Examining Generalized Anxiety Disorder Among Healthcare Workers During the COVID-19 Pandemic Within the Democratic Republic of the Congo' has been provisionally accepted for publication in PLOS Mental Health.

Best regards,

Karli Montague-Cardoso

Executive Editor

PLOS Mental Health

Reviewer Comments (if any, and for reference):

Reviewer's Responses to Questions

**Comments to the Author**

1. If the authors have adequately addressed your comments raised in a previous round of review and you feel that this manuscript is now acceptable for publication, you may indicate that here to bypass the “Comments to the Author” section, enter your conflict of interest statement in the “Confidential to Editor” section, and submit your "Accept" recommendation.

Reviewer #1: All comments have been addressed

Reviewer #2: All comments have been addressed

2. Does this manuscript meet PLOS Mental Health’s publication criteria ? Is the manuscript technically sound, and do the data support the conclusions? The manuscript must describe methodologically and ethically rigorous research with conclusions that are appropriately drawn based on the data presented.

Reviewer #1: Yes

Reviewer #2: Yes

3. Has the statistical analysis been performed appropriately and rigorously?

Reviewer #1: Yes

Reviewer #2: Yes

4. Have the authors made all data underlying the findings in their manuscript fully available (please refer to the Data Availability Statement at the start of the manuscript PDF file)?

Reviewer #1: Yes

Reviewer #2: Yes

5. Is the manuscript presented in an intelligible fashion and written in standard English?

Reviewer #1: Yes

Reviewer #2: Yes

6. Review Comments to the Author

Reviewer #1: Thank you for addressing all my concerns. I have no further comments. I hope this has helped to improve the quality of the manuscript, which I believe will contribute to the advancement of knowledge in the field of mental health.

Reviewer #2: All my concerns have been met by the author

7. PLOS authors have the option to publish the peer review history of their article (what does this mean? ). If published, this will include your full peer review and any attached files.

**Do you want your identity to be public for this peer review?** For information about this choice, including consent withdrawal, please see our Privacy Policy .

Reviewer #1: **Yes: ** Erick Mukala Mayoyo

Reviewer #2: **Yes: ** Dr David Onchonga
